# Genome editing in butterflies reveals that *spalt* promotes and *Distal-less* represses eyespot colour patterns

Linlin Zhang[1] & Robert D. Reed[1]

Butterfly eyespot colour patterns are a key example of how a novel trait can appear in association with the co-option of developmental patterning genes. Little is known, however, about how, or even whether, co-opted genes function in eyespot development. Here we use CRISPR/Cas9 genome editing to determine the roles of two co-opted transcription factors that are expressed during early eyespot determination. We found that deletions in a single gene, *spalt*, are sufficient to reduce or completely delete eyespot colour patterns, thus demonstrating a positive regulatory role for this gene in eyespot determination. Conversely, and contrary to previous predictions, deletions in *Distal-less* (*Dll*) result in an increase in the size and number of eyespots, illustrating a repressive role for this gene in eyespot development. Altogether our results show that the presence, absence and shape of butterfly eyespots can be controlled by the activity of two co-opted transcription factors.

[1] Department of Ecology and Evolutionary Biology, Cornell University, 215 Tower Road, Ithaca, New York 14853-7202, USA. Correspondence and requests for materials should be addressed to R.D.R. (email: robertreed@cornell.edu).

The eyespot patterns found on butterfly wings are among the most striking colour patterns in the natural world, and there is a large literature concerning their ecology, evolution and development[1,2]. Across different species eyespots play a number of ecological roles, including in such disparate phenomena as mate choice and predation avoidance, and thus provide a powerful example of how a single trait can adapt to serve many different functions under different types of selective pressures. In 1994, Carroll *et al.* found that the transcription factor gene *Dll*, which plays an ancestral role in animal appendage formation[3], is expressed in early eyespot determination[4]. This discovery provided one of the most surprising and marked examples of evolutionary gene co-option, that is, the redeployment of an ancestral gene for a novel function[5], and remains among the most widely cited case studies in evolutionary developmental biology. Subsequent work has identified a number of other apparently co-opted genes expressed at different times during eyespot development, some of which show compelling phylogenetic patterns of expression gain-and-loss that suggest evolutionary dynamicity in the eyespot regulatory network[1,6,7]. A major challenge in eyespot development work, however, has been a lack of functional data. Although candidate gene expression patterns are clearly associated with eyespots both developmentally and evolutionarily, we still do not know how, or even whether, these genes function in colour pattern formation. Because of this it has not been possible to assess the morphological or adaptive significance of eyespot-associated gene co-option.

Here we address to what extent co-opted genes expressed during early butterfly eyespot determination function in eyespot development. Comparative expression studies show that the transcription factor spalt is one of the earliest factors expressed in eyespot foci, and phylogenetic work suggests that this early spalt expression is a conserved ancestral trait across nymphalid butterflies[6,8]. Because of these observations it has been proposed that spalt may be a key upstream regulator of eyespot determination[1]. Another transcription factor expressed in association with eyespots across many species is Dll[6,9]. Various lines of evidence, primarily comparative expression studies, but also including some overexpression and sequence association work[10–12], have led to the hypothesis that Dll is also a positive regulator of eyespots[1]. Here we use CRISPR/Cas9 genome editing to assess the developmental functions of spalt and Dll in eyespot colour pattern development, and, more broadly, to show that the co-option of these transcription factors has played a role in the evolution of butterfly wing patterns.

## Results

**Pigmentation gene deletion demonstrates wing mosaics.** To test the function of candidate eyespot determination genes we opted for a loss-of-function approach using CRISPR/Cas9 targeted deletions[13,14]. To test the utility of CRISPR/Cas9 for making $G_0$ deletion mosaic butterflies, we optimized a protocol in the painted lady butterfly *Vanessa cardui* by targeting the melanin pigmentation gene *Dopa decarboxylase* (*Ddc*), reasoning that pigmentation defects should allow for easy visualization of potential mosaicism in wings (Fig. 1). On the basis of genotyping a panel of 81 embryos we found that our optimized double-guide RNA method resulted in a 69% rate of long deletions at the target locus. Furthermore, across all deletion clones, including for *spalt* and *Dll* as mentioned below, we observed a 66% frameshift rate downstream of deletions. Consistent with *Ddc* phenotypes observed in *Drosophila melanogaster* mutants, the majority of injected animals that survived to hatching age were unable to emerge from their eggs

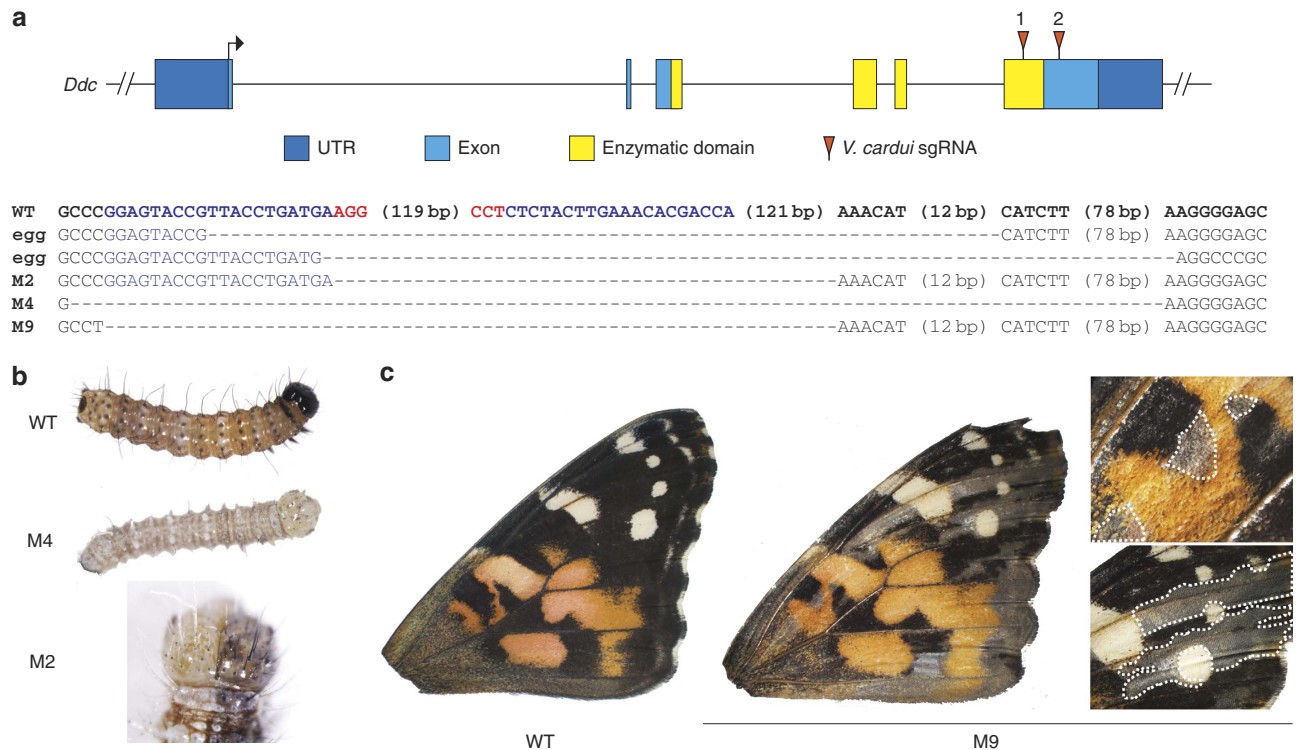

**Figure 1 | CRISPR/Cas9 deletions in the melanin pigmentation gene *Ddc* result in mosaic depigmentation phenotypes in *V. cardui*.** (**a**) Location of sgRNAs relative to the *V. cardui Ddc* locus. Sequences of selected deletion alleles from eggs and the butterflies shown (M2, M4 and M9) confirm disruption of *Ddc*. Blue: sgRNA targets. Red: PAM sequences. (**b**) *Ddc* deletions result in depigmentation of larvae, including bilateral mosaics. (**c**) An example of mosaic melanin-specific depigmentation in adult wings.

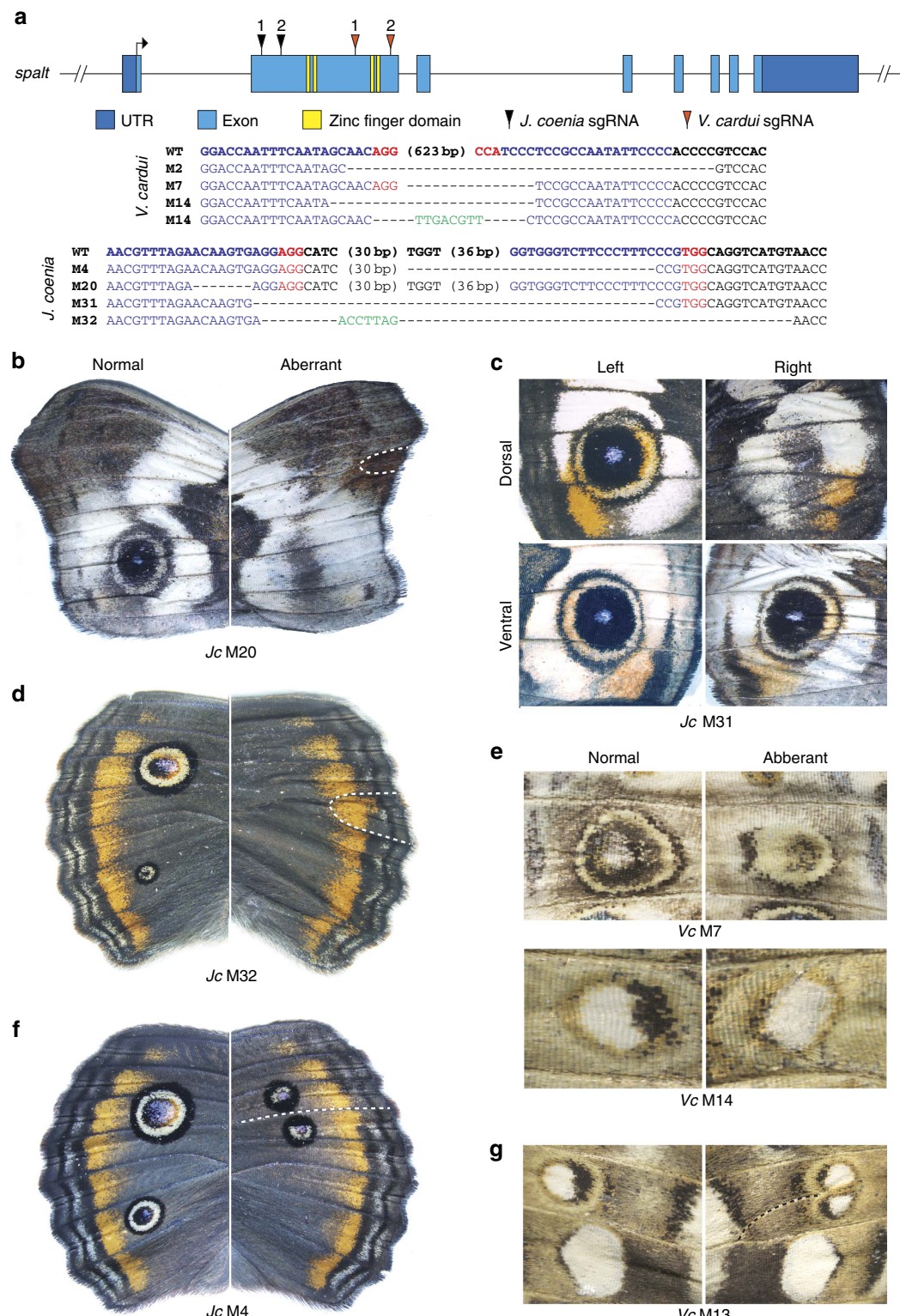

**Figure 2 | CRISPR/Cas9 *spalt* deletions result in reduction and loss of eyespot colour patterns.** (**a**) Location of sgRNAs relative to the *V. cardui spalt* locus. Sequences of *spalt* alleles from the animals shown confirm lesions in target regions. Blue: sgRNA targets. Red: PAM sequences. Green: novel sequences not observed in wild-type alleles. Somatic deletions in *spalt* cause complete loss of forewing eyespots in *J. coenia*: (**b**) ventral forewing; (**c**) dorsal forewing; (**d**) dorsal hindwing. (**e**) *spalt* deletions reduce eyespots in *V. cardui* ventral hindwings (top) and forewings (bottom). Ectopic wing veins (dashed lines) resulting from *spalt* deletion subdivide eyespot patterns in (**f**) *J. coenia* (also showing a missing posterior eyespot), and (**g**) *V. cardui*. All comparisons shown are left–right asymmetrical phenotypes from individual injected butterflies. 'Normal' phenotypes are as wild-type, 'aberrant' phenotypes are non-wild type and vary asymmetrically from the normal patterns on the same animal.

without assistance, likely due to deficiencies in mouthpart sclerotization[15]. Larvae showed mosaic pigmentation defects, including bilateral mosaics, which ranged in severity (Fig. 1b), and adult wings were mosaic for melanin-deficient clones (Fig. 1c). In sum, by using a pigment marker gene we demonstrate that CRISPR/Cas9 can be used to produce butterflies with adult wings that are mosaic for targeted deletions. This ability to produce somatic deletion mosaics is an important new tool in the butterfly experimental system, as it permits the analysis of tissue-specific deletion effects that would otherwise be embryonic lethal in pure mutant lines.

**Spalt is a positive regulator of eyespot colour patterns**. To test the hypothesis that *spalt* is a regulator of eyespot colour patterns, we used CRISPR/Cas9 to produce deletions, and related frameshifts, in the *spalt* coding region of two nymphalid species that display eyespots (Fig. 2a): *Junonia coenia* and *V. cardui*. In both the species inducing deletions in *spalt* was sufficient to produce aberrant mosaic-wing phenotypes not seen in natural populations (Table 1). Of greatest interest was the frequent reduction and/or complete loss of eyespot colour patterns in wings of injected animals of both species (Fig. 2, Supplementary Fig. 1). This effect was particularly marked in *J. coenia* where *spalt* deletions produced many individuals completely missing eyespots from the forewing and/or hindwing (Fig. 2b–d,f). We also noted that the loss of an eyespot on one wing surface (that is, dorsal or ventral) could occur without affecting an eyespot on the opposing surface (Fig. 2c), demonstrating that even though eyespots occur in the same position on opposing wing surfaces, their determination is dorsoventrally decoupled to a large degree.

We did not observe the same type of complete eyespot loss in *V. cardui* as we saw in *J. coenia*, possibly due to different deletion positions (*V. cardui* deletion alleles retained two unaffected 5′ zinc finger domains relative to *J. coenia*). Nonetheless, we still observed many clear cases of mosaic eyespot reduction in *V. cardui* (Fig. 2e, Supplementary Fig. 1). In all cases in both species, eyespot diminution due to *spalt* deletion had a highly specific effect only on eyespot patterns, leaving neighbouring colour patterns unaffected. In addition to eyespot reduction phenotypes, *spalt* deletions in both *J. coenia* and *V. cardui* frequently resulted ectopic wing veins (Fig. 2b,d,f,g), consistent with *spalt* mutant phenotypes in *D. melanogaster*[16]. Interestingly, these ectopic veins often traced through the middle of eyespots, dividing eyespots into two (Fig. 2f,g). In *J. coenia*, we also observed several cases where vein abnormalities around the discal cell were associated with reduction of discal spot patterns (Supplementary Fig. 1); however, we tentatively speculate that these colour pattern defects are due primarily to vein-related

pattern disruption since *spalt* expression has not been observed in discal spot patterns. Altogether our results show that *spalt* is a positive regulator of eyespots, and that it is required for eyespot determination in *J. coenia*.

**Dll is an eyespot repressor and organizes distal colour patterns.** To test the hypothesis that *Dll* is an activator of eyespot determination we used CRISPR/Cas9 to induce deletions, and related frameshifts, in the *Dll* coding region in *J. coenia* and *V. cardui* (Fig. 3a). Consistent with the known function of *Dll* in *D. melanogaster* appendage development[17], we observed a high frequency of missing or reduced legs, antennae, and labial palps in injected animals of both species (Supplementary Fig. 2, Table 1). The resulting eyespot phenotypes were surprising, however, because *Dll* deletions had marked positive effects on both eyespot size and number. In both *J. coenia* and *V. cardui*, *Dll* deletions were sufficient to promote spatial expansion of eyespots towards the wing margin, causing the eyespots to be distally elongated, and, consequently, larger (Fig. 3b–d). These eyespot elongation phenotypes were reminiscent of *Bicyclus anynana comet* mutants[18], as well as natural eyespot phenotypes seen in some other butterflies such *Chloreuptychia spp.*, suggesting that *Dll* downregulation may be implicated in these other phenotypes as well. Overall, we saw the most extreme *Dll* phenotypes in *V. cardui*, in which we recovered individuals with rows of multiple novel, ectopic eyespots on the forewing (Fig. 3b). These ectopic spots were distally expanded, as was the case for all aberrant eyespots resulting from *Dll* deletion.

In addition to changes in shape, size and number of eyespots, *Dll* deletions also resulted in several other colour pattern defects (Table 1), including the disruption and loss of marginal bands and parafocal elements in both species (Fig. 3b–e, Supplementary Fig. 2), loss of pigmentation in *V. cardui* eyespot foci (Fig. 3c), and the appearance of hyperpigmented clones across the *J. coenia* wing surface (Fig. 3d,e). It was notable that *Dll* deletions had a different range of wing pattern effects in *V. cardui* versus *J. coenia* despite the fact that the efficiency of the *Dll* mutagenesis appeared to be similar, as evidenced by similar rates of appendage mutations (52% in *V. cardui* and 41% in *J. coenia*; Table 1). For instance in *J. coenia* we did not observe the marked gain-of-eyespot phenotypes observed in *V. cardui*, but conversely, *V. cardui* mutants did not show hyperpigmented clones as seen in *J. coenia*. This leads us to speculate that *Dll* may play some divergent colour patterning functions in the two genera we examined, in addition to its shared functions in eyespot and margin pattern determination. This idea, however, is presented with the caveat that we cannot rigorously confirm lack-of-function (or cell autonomy) without a clone boundary

**Table 1 | Summary of CRISPR/Cas9 injection results.**

*spalt*

| Species | Injected eggs | Hatched eggs | Hatch ratio | Total adults | Mutant phenotypes | Eyespot reduction | Eyespot loss | Abnormal veins | Vein-related discal spot distortion |
|---|---|---|---|---|---|---|---|---|---|
| *V. cardui* | 421 | 281 | 66.27% | 25 | 14 (56%) | 11 | 1 | 3 | 0 |
| *J. coenia* | 508 | 90 | 17.71% | 63 | 21 (33.3%) | 16 | 20 | 8 | 8 |

*Distal-less*

| Species | Injected eggs | Hatched eggs | Hatch ratio | Total adults | Mutant phenotypes | Antenna, leg, or palp reduction | Eyespot expansion | Additional eyespots | White HW eyespot foci | Parafocal and margin distortion | Dark patches |
|---|---|---|---|---|---|---|---|---|---|---|---|
| *V. cardui* | 772 | 609 | 78.80% | 85 | 44 (51.7%) | 44 | 16 | 16 | 5 | 16 | 0 |
| *J. coenia* | 323 | 129 | 38.15% | 95 | 39 (41.0%) | 39 | 1 | 0 | 0 | 4 | 3 |

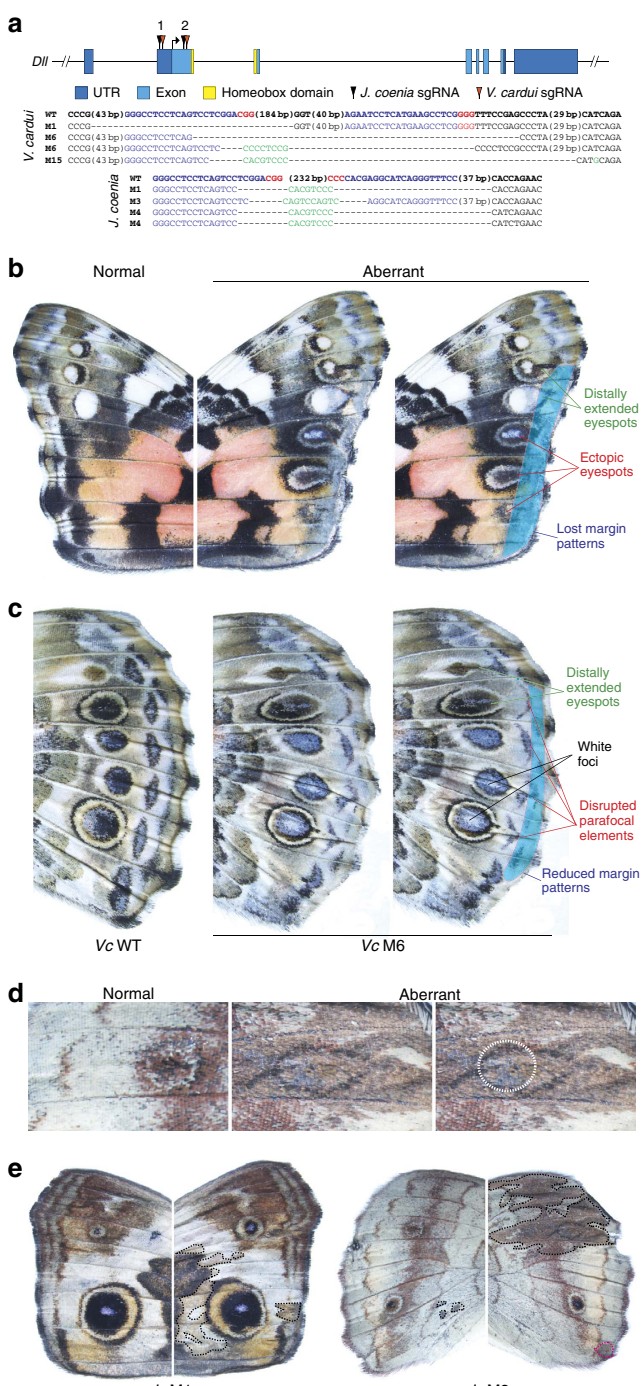

**Figure 3 | CRISPR/Cas9 *Dll* deletions result in expansion and gain of eyespot colour patterns.** (**a**) Location of sgRNAs relative to the *V. cardui Dll* locus. Sequences of *Dll* alleles from the animals shown confirm lesions in target regions. Blue: sgRNA targets. Red: PAM sequences. Green: novel sequences not observed in wild-type alleles. (**b**) *Dll* deletion causes multiple effects on distal colour patterns of the *V. cardui* forewing, including an increase in size and number of eyespots. Both wings shown are from the same individual (*Vc* M15), where the 'normal' wing shows a wild-type phenotype, while the 'aberrant' wing shows several abnormalities as annotated to the right. (**c**) Effects of *Dll* mutation on the *V. cardui* hindwing. (**d**) A *J. coenia Dll* mutant (*Jc* M3) shows distal elongation of a hindwing eyespot. The white dotted circle shows the size of the wild-type eyespot on the opposing wing of the same individual. (**e**) *J. coenia Dll* deletion result in patches of dark pigmentation on both the forewing (left) and hindwing (right). The individual on the left also shows disruption of margin patterns and anterior eyespot size asymmetry.

marker, as is standard in *D. melanogaster* work[19]. Nonetheless, our strong *Dll* deletion phenotypes in both species clearly show that *Dll* plays a variety of roles in colour pattern development, including shaping and repressing eyespots, organizing parafocal and margin colour patterns, and regulating pigmentation.

## Discussion

In this study we used CRISPR/Cas9 somatic mosaics to show that the transcription factor genes *spalt* and *Dll* play key roles in butterfly wing pattern development, and by extension, that the co-option of these genes has played a role in colour pattern evolution. The effect of *spalt* deletion—that is, the loss and/or reduction of eyespots—clearly shows that this gene plays a role in promoting eyespots. *spalt* is one of the earliest known factors to presage eyespots in imaginal disc development; therefore, it is reasonable to hypothesize that this gene is a positive upstream regulator of eyespot determination, as is consistent with earlier models based on expression associations.

*Dll* deletion phenotypes, on the other hand, were both varied and surprising. Perhaps the greatest surprise was finding that *Dll* acts as a repressor of eyespot patterns, contrary to expectations arising from decades of comparative expression work. Despite observing many strong *Dll* deletion phenotypes that showed clear eyespot effects, we saw no evidence in our experiments that *Dll* deletion had any diminishing effects on eyespots. All aberrant eyespot phenotypes we observed involved an increase in eyespot size and/or number. We are now challenged to reconcile this repressive function with previous expression studies that show a positive association between *Dll* expression and the presence and size of eyespots. In this regard it is likely important that in *V. cardui* and *J. coenia* expression of early eyespot-associated factors, including spalt, occurs in eyespot foci before Dll, at a time when only fine, transient lines of Dll expression extend from the eyespot foci to the wing margin[6,9] (for example, Fig. 4). If these lines of Dll expression play an early role to inhibit distal expression of eyespot determination genes, then loss of these Dll lines could result in extension of eyespot-activating gene expression towards the margin, as is consistent with our observed phenotypes. We therefore speculate that a primary colour patterning role of *Dll* is to inhibit eyespot determination genes in the distal wing margin during an early stage of pattern formation, before *Dll* activation in eyespot foci (for example, Fig. 4c). It is still unclear, however, why some spots are completely repressed by *Dll* (for example, *V. cardui* posterior forewing spots), while others are only partially repressed (for example, *V. cardui* anterior forewing spots)—further work will be required to address this. It is also unclear why *Dll* is expressed in imaginal disc eyespot foci in the species we examined, since our results do not support a role for *Dll* in focal determination. We speculate that *Dll* focal expression may instead play a later role in pigmentation, as implied by the loss of pigmentation in *V. cardui* hindwing eyespot foci in response to *Dll* deletion (Fig. 3c).

The effects of *Dll* deletion on non-eyespot traits including parafocal and margin colour pattern elements, and pigmentation, were surprising and of significant interest. Although previous studies have not focused much attention on the possible role of *Dll* in the development of these other colour pattern traits, these effects are in line with earlier observations. Importantly, strong wing margin expression of *Dll* (Fig. 4a) is consistent with an organizing role for marginal and parafocal colour patterns[9]. Indeed, a developmental link between eyespots, marginal bands and parafocal elements is observed in coldshock experiments in *V. cardui* that cause eyespots and parafocal elements to merge and marginal bands to expand[20]. We further speculate that the

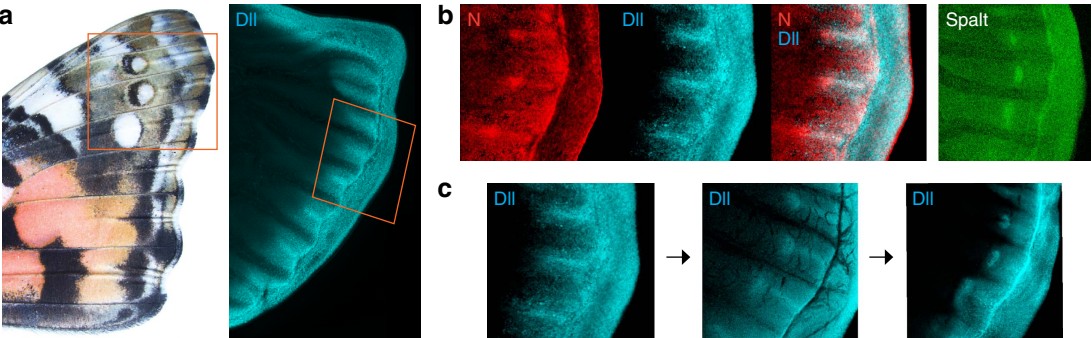

**Figure 4 | Eyespot determination precedes Dll focal expression. (a)** *Dll* expression in line patterns in a mid-last instar *V. cardui* forewing imaginal disc. The red box highlights the area of the wing shown in subsequent panels. (**b**) Eyespot markers Notch and spalt are expressed in foci at a time point when Dll is only expressed in a line extending from the eyespot focus to the wing margin. (**c**) A time series of Dll expression shows the progression of line, to line-plus-focus (corresponding to **b**), to focus only.

pigment-altering effects of *Dll* mutagenesis are likely related to pupal-stage Dll expression in eyespot foci and more widely across the wing epithelium[21]. Pigment regulation by *Dll* is also supported by work in *B. anyana* and *Junonia orithya* showing that ectopic expression of Dll causes localized changes in pigmentation, but is insufficient to induce distinct patterns elements[10,12]. Future models of wing pattern development must consider the complex role of Dll as both a promoter and repressor of multiple colour pattern traits, especially along the distal wing margin.

Altogether, our results show that the early stages of butterfly eyespot determination involve the activity of both promoting and repressing regulators, and that the presence, absence, shape and colour of eyespots can be modulated by the activity of two co-opted genes. Our surprising finding that *Dll* is a repressor of eyespot determination requires a significant revision of current models of eyespot development and provides a cautionary tale about interpreting correlations between traits and candidate gene expression levels. In addition to providing the first unambiguous functional validation of the role of specific genes in butterfly wing pattern determination, our study also demonstrates the power and potential of CRISPR/Cas9 somatic mosaics for work in new model systems.

## Methods

**Animals.** *V. cardui* caterpillars were obtained from Carolina Biological Supply (Burlington, NC, USA) and *J. coenia* eggs were obtained from Fred Nijhout (Duke University, NC, USA). All butterflies were reared on artificial diet at 16/8-h light/darkness at 28 °C and 70% humidity.

**Preparation of guide-RNAs.** Our strategy for producing long deletions in target loci was to generate two Cas9 cut sites flanking the desired lesion interval using two single guide-RNAs (sgRNAs), then to rely on nonhomologous end joining to close the resulting gap[22]. sgRNAs were designed by manually searching genomic regions for $GGN_{18}NGG$ or $N_{20}NGG$ protospacer adjacent motif (PAM) sequences on the sense or antisense strands (Supplementary Table 1). sgRNA template was generated by PCR using Phusion polymerase (New England Biolabs, Ipswich, MA, USA). *In vitro* transcription was conducted using Megascript T7 Kit (Ambion, Waltham, MA, USA) and purified by phenol–chloroform extraction and isopropanol precipitation.

**Microinjection of butterfly embryos.** Butterfly eggs were collected from host plant leaves within an hour of being laid and arranged on double-sided adhesive tape on a microscope slide. To soften the egg chorion *J. coenia* eggs were dipped in 5% benzalkonium chloride (Sigma-Aldrich, St Louis, MO, USA) for 90 s, and then washed in water for 2 min before mounting on microscope slide. Eggs were then dried for 15 min in a desiccant chamber. 0.5 μg of Cas9 protein (PNA Bio Inc., Thousand Oaks, CA, USA) and 250 ng of each sgRNA were mixed in a 2.5 μl volume and injected into eggs using a 0.5-mm borosilicate needle (Sutter Instruments, Novato, CA, USA).

**Mutation genotyping and trait scoring.** For *Ddc* larval genotyping genomic DNA was extracted from caterpillars using proteinase K in digestion buffer[13]. For all other genotyping DNA was extracted from wing muscle tissue using a QIAamp

DNA mini kit (Qiagen, Hilden, Germany). Fragments flanking the CRISPR target regions were amplified by PCR, gel-purified, subcloned into a TOPO TA vector (Invitrogen, Carlsbad, CA, USA), and sequenced on an ABI 3730 sequencer. Genotyping primers are provided in Supplementary Table 1. To rule out non-CRISPR variation when scoring phenotypes, we only called an individual as having an aberrant deletion phenotype if a trait showed a strong left–right asymmetry. Although there is a small degree of individual wing pattern variation in species we surveyed, this variation is typically symmetrical. We observed asymmetrical, mosaic variation only in individuals injected with sgRNAs and Cas9.

**Immunohistochemistry.** Larval wing discs were dissected, staged, fixed, stained and imaged as previously described[8,21] using a rabbit anti-spalt polyclonal antibody[23] at a 1:200 dilution, a rabbit anti-Dll polyclonal antibody[24] at a 1:100 dilution, and/or a mouse anti-Notch monoclonal antibody (C17.9C6)[25] at a 1:200 dilution.

**Data availability.** Images of all wings from butterflies showing presumptive deletion phenotypes. along with a reference spreadsheet, are available on Dryad: http://dx.doi.org/10.5061/dryad.tj45p.

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

## Acknowledgements

This work was supported by National Science Foundation grant DEB-1354318 to R.D.R. We thank R. Barrio for spalt antibody, G. Panganiban for Dll antibody, the University of Iowa Developmental Studies Hybridoma Bank for Notch antibody and F. Nijhout for *J. coenia* eggs. We thank J. Lewis, A. Martin, A. Mazo-Vargas, M. Sheehan, K. van der Burg and M. Wolfner for discussions and comments on the manuscript.

## Author contributions

R.D.R. and L.Z.: designed the study and wrote the paper. L.Z.: CRISPR/Cas9 genome editing. R.D.R.: immunohistochemistry.

## Additional information

**Competing financial interests:** The authors declare no competing financial interests.

