## [Peer Review File · Nature Communications]

REVIEWERS' COMMENTS:

Reviewer #1 (Remarks to the Author):

This paper uses CRISPR gene editing to study wing patterning genes in butterfly eyespots. Although not the first gene editing or CRISPR paper in butterflies, it is the first to offer significant biological insights into wing pattern determination, so I do consider this something of a landmark paper which will set the direction of the field for the foreseeable future. The results are quite surprising given previous data on these genes and suggests many more surprises are in store once we investigate other wing patterning genes with this approach.

My major request is that the authors provide full disclosure of the results of the experiments. Right now the phenotypes generated are illustrated by a single image - for example a key result is the production of ectopic eyespots in *Junonia* in response to Dll knockouts, which is illustrated by a single image in Figure 3. Table S1 is a helpful summary of the results but it would be more transparent if these results could be assessed more directly by the reader. I would like to see images of all of the mutant butterflies - it should be relatively straight forward to provide these as supplementary materials.

This is a minor point - I would hesitate to suggest that the Dll mutants are atavistic ancestral phenotypes - there have been many gains and losses of particular eyespots across the butterflies which may or may not have involved Dll repression. Therefore this is a bit reminiscent of suggestions that *Ubx* fly mutants produce an ancestral four-winged fly, which is obviously a naive interpretation of this phenotype.

Otherwise the paper is clearly written and I think the conclusions and discussion is justified. I strongly recommend publication.

Reviewer #2 (Remarks to the Author):

Review of "Genetic basis of butterfly eyespot determination" by Zhang and Reed.

This is a well put together & timely paper. It should be published in Nature Communications. In my opinion it would easily be in the top 25% of papers published in this journal in terms of impact.

The paper using CRISPR to knock out 3 genes in two butterfly species. One of those species is the common painted lady that many of us have seen in our back yards. One of the genes is a control, and the other two are developmental genes (*spalt* and *Distalless*) believed to be important in eyespot patterns on butterfly wings. I imagine this paper could be published strictly for technical reasons, namely getting Crispr to work in this system, but there is a little more to the story.

These two genes have been studied to death in butterflies. There are many papers that have looked at expression patterns in wing imaginal discs (using either antibodies or mRNA in situ). Based on these data the consensus in the field is that Dll defines eyespot foci, and hence over-expression of Dll will result in larger or extra eyespots. Instead, the authors show here that Dll likely acts to repress eyespot growth - the exact opposite conclusion. Presumably earlier studies were looking at Dll expression a little later in development after its repressive role had ended. So this paper illustrates the value of knocking out these genes as opposed to just looking at expression patterns.

An interesting aspect of the paper is that the authors do not make complete and/or heritable gene knockouts (KOs). Instead they look at individuals where the KO is somatic and hence any given study individual a mosaic. They infer mosaicism by looking for individuals where the left wing appears WT and the right mutant (or the opposite), and have great images showing this. Being able to study mosaics is powerful, since null of these genes are likely homozygous lethal (and in the case of Dll, I believe haplo-insufficient).

I do have two small concerns with the paper.

First, panel A of Figures 1,2,&3 are confusing (or at least not the way Drosophila papers present things - see any of the Gratz papers for example). Is what is being depicted the intron/exon structure of the gene with "blue boxes" exons ... or are the authors depicting a cDNA with functional domains being highlighted. For my money, I would like to see the gene structure, as alternative splicing could make some CRISPR hits not nulls. It would be nice to know how the lesions identified actually relate to their associated gene. This should certainly be further clarified, perhaps as a supplement.

Second, the authors have to be a little careful with their mosaic analysis. It is not without pitfalls. Historically in flies mosaic studies have used mitotic recombination to make mosaics, with patches homozygous for a mutant allele also marked with a second mutant. So for example, if I was studying Dpp on the wing - I might use "forked" as a marker. So wing cells with a forked bristle are inferred to be Dpp[-]. In this paper Dll/spalt are both the marker and the gene about which inference is being made - and this potentially creates some problems with inference. Perhaps the issue is subtle, if we are really confident that these genes are cell autonomous, but this limitation of the method should be discussed. In this sense a better experiment might have knocked-in some dominant marker that could be scored on the wings - so the region null for Dll/spalt was marked independently of these genes themselves. I in Drosophila not doing the experiment this way would be perceived of as a major problem ... but the authors should given a little slack given the system.

Reviewer #3 (Remarks to the Author):

The dramatic and diverse colour patterns on butterfly wings are a traditional, and currently very active, system for the study of evolution and evolutionary development. The present work is both important and novel, using the CRISPR/Cas method to disable two genes and thereby to clearly demonstrate their different functions in the development of eyespot patterns within two species.

I have a few comments:

I feel that the Methods and the figure legends [to parts (a) of Figs 1-3] could be rather more informative about the method - eg explaining the abbreviations 'sg RNA' and 'PAM' sequences, and the significance of the blue box sections of the proteins (and surely not 'HOX' for Distal-less!).

The findings for spalt are clear and striking and they indicate positive regulation of eyespot formation and size, in accordance with expectations based on the larval expression of the gene in eyespot foci.

Since the illustrations in Fig 2 contrast the normal and aberrant (assumed mosaic mutant) wings of the same animal, a clearer wording in sentence 3 of paragraph 1 of this section, would be '.....completely missing eyespots from the forewing and/or the hindwing...'

In the legend for Fig 2, part (c) shows both the dorsal and ventral forewing.

spalt disruption also produces extra sections of wing vein which can result in splitting of an eyespot (not unexpected from previous work). Table S2 also lists, in Junonia, 'vein-related discal spot distortion' but these (6 cases) are not mentioned - is this spot similarly split by extra vein material - this is intriguing as the discal spot is in a very different position on the wing and does it have central expression of spalt ?

The dramatic Vanessa findings for distal-less clearly indicate a contrasting repressive function in eyespot development, despite its larval expression in the foci but not in corresponding positions in other wing-cells.

In the legend to Fig 3 part (b) the aberrant wing is said to show a 'strong deletion phenotype', which is rather odd wording!

The effects on Junonia are much weaker, showing no additional eyespots, few marginal effects and only one case of eyespot expansion (Fig 3 d). However, Fig 3e shows specimen JcM1 as an illustration of background pigmentation effects - does this wing not also show a major enlargement of the anterior eyespot (judging from the other wing and from Fig 2b)?

These results are striking, somewhat variable between the two species and very surprising in the case of distal-less - all of these aspects are clearly and interestingly discussed in the paper.

This study is important and the intriguing results will certainly stimulate further work. The manuscript should certainly be published.

REVIEWERS' COMMENTS:

Reviewer #1 (Remarks to the Author):

My earlier comments were fairly minor so I am happy with the manner in which the authors have responded to all comments. It is great that images are now available as supplementary material, however I was unable to match up the images with the data in Table S1 - the mutant IDs in the table did not seem to be associated with the images. This data needs to be better documented (there may be something I have missed, or it may be something about the way the images are presented on the website, but I suggest writing the individual IDs into the images in photoshop so they can be identified - otherwise its a bit pointless having these images available)

Otherwise fine to go ahead, I recommend publication

[Authors' response in bold]

Reviewer #1 (Remarks to the Author):

This paper uses CRISPR gene editing to study wing patterning genes in butterfly eyespots. Although not the first gene editing or CRISPR paper in butterflies, it is the first to offer significant biological insights into wing pattern determination, so I do consider this something of a landmark paper which will set the direction of the field for the foreseeable future. The results are quite surprising given previous data on these genes and suggests many more surprises are in store once we investigate other wing patterning genes with this approach.

My major request is that the authors provide full disclosure of the results of the experiments. Right now the phenotypes generated are illustrated by a single image - for example a key result is the production of ectopic eyespots in *Junonia* in response to *Dll* knockouts, which is illustrated by a single image in Figure 3. Table S1 is a helpful summary of the results but it would be more transparent if these results could be assessed more directly by the reader. I would like to see images of all of the mutant butterflies - it should be relatively straight forward to provide these as supplementary materials.

- **We agree that it would be a great idea to make pictures available of all mutant phenotypes. This collection of images, however, represents 2.5 GB of raw data – too much for a supplementary figure. We will therefore upload the entire high-resolution image dataset to DRYAD, and include a new supplementary table (Table S1) so readers can easily locate pictures of individual butterflies with specific phenotypes. Note also that Figures S1 and S2 show several representative examples of more minor phenotypes.**

This is a minor point - I would hesitate to suggest that the *Dll* mutants are atavistic ancestral phenotypes - there have been many gains and losses of particular eyespots across the butterflies which may or may not have involved *Dll* repression. Therefore this is a bit reminiscent of suggestions that *Ubx* fly mutants produce an ancestral four-winged fly, which is obviously a naive interpretation of this phenotype.

- **We have removed the atavism discussion.**

Otherwise the paper is clearly written and I think the conclusions and discussion is justified. I strongly recommend publication.

Reviewer #2 (Remarks to the Author):

Review of "Genetic basis of butterfly eyespot determination" by Zhang and Reed.

This is a well put together & timely paper. It should be published in Nature Communications. In my opinion it would easily be in the top 25% of papers published in this journal in terms of impact.

The paper using CRISPR to knock out 3 genes in two butterfly species. One of those species is the common painted lady that many of us have seen in our back yards. One of the genes is a control, and the other two are developmental genes (spalt and Distalless) believed to be important in eyespot patterns on butterfly wings. I imagine this paper could be published strictly for technical reasons, namely getting Crispr to work in this system, but there is a little more to the story.

These two genes have been studied to death in butterflies. There are many papers that have looked at expression patterns in wing imaginal discs (using either antibodies or mRNA in situs). Based on these data the consensus in the field is that Dll defines eyespot foci, and hence over-expression of Dll will result in larger or extra eyespots. Instead, the authors show here that Dll likely acts to repress eyespot growth - the exact opposite conclusion. Presumably earlier studies were looking at Dll expression a little later in development after its repressive role had ended. So this paper illustrates the value of knocking out these genes as opposed to just looking at expression patterns.

An interesting aspect of the paper is that the authors do not make complete and/or heritable gene knockouts (KOs). Instead they look at individuals where the KO is somatic and hence any given study individual a mosaic. They infer mosaicism by looking for individuals where the left wing appears WT and the right mutant (or the opposite), and have great images showing this. Being able to study mosaics is powerful, since null of these genes are likely homozygous lethal (and in the case of Dll, I believe haplo-insufficient).

I do have two small concerns with the paper.

First, panel A of Figures 1,2,&3 are confusing (or at least not the way Drosophila papers present things - see any of the Gratz papers for example). Is what is being depicted the intron/exon

structure of the gene with "blue boxes" exons ... or are the authors depicting a cDNA with functional domains being highlighted. For my money, I would like to see the gene structure, as alternative splicing could make some CRISPR hits not nulls. It would be nice to know how the lesions identified actually relate to their associated gene. This should certainly be further clarified, perhaps as a supplement.

- **We have revised our figures to show detailed genomic annotations of the *Ddc*, *spalt*, and *Distal-less* loci.**

Second, the authors have to be a little careful with their mosaic analysis. It is not without pitfalls. Historically in flies mosaic studies have used mitotic recombination to make mosaics, with patches homozygous for a mutant allele also marked with a second mutant. So for example, if I was studying Dpp on the wing - I might use "forked" as a marker. So wing cells with a forked bristle are inferred to be Dpp[-]. In this paper Dll/spalt are both the marker and the gene about which inference is being made - and this potentially creates some problems with inference. Perhaps the issue is subtle, if we are really confident that these genes are cell autonomous, but this limitation of the method should be discussed. In this sense a better experiment might have knocked-in some dominant marker that could be scored on the wings - so the region null for Dll/spalt was marked independently of these genes themselves. I in *Drosophila* not doing the experiment this way would be perceived of as a major problem ... but the authors should give a little slack given the system.

- **We agree, and we are pleased that the reviewer is willing to grant us some latitude in this case. Fortunately, all reviewers agree that the exceptionally strong mosaic phenotypes in our study are sufficient to support our conclusions. In order to clarify the limitations of our approach for readers, we added the following text to Page 4: "This idea, however, is presented with the caveat that we cannot rigorously confirm lack-of-function (or cell autonomy) without a clone boundary marker, as is standard in *D. melanogaster* work¹⁹."**

Reviewer #3 (Remarks to the Author):

The dramatic and diverse colour patterns on butterfly wings are a traditional, and currently very active, system for the study of evolution and evolutionary development. The present work is both important and novel, using the CRISPR/Cas method to disable two genes and thereby to clearly demonstrate their different functions in the development of eyespot patterns within two species.

I have a few comments:

I feel that the Methods and the figure legends [to parts (a) of Figs 1-3] could be rather more informative about the method - eg explaining the abbreviations 'sg RNA' and 'PAM' sequences, and the significance of the blue box sections of the proteins (and surely not 'HOX' for Distal-less!).

- **We added more detail and a new reference to the Methods to better describe our double-cleavage nonhomologous end joining repair strategy, and we now clearly define the “sgRNA” and “PAM” abbreviations. We also improved our locus annotations in the figures.**

The findings for *spalt* are clear and striking and they indicate positive regulation of eyespot formation and size, in accordance with expectations based on the larval expression of the gene in eyespot foci. Since the illustrations in Fig 2 contrast the normal and aberrant (assumed mosaic mutant) wings of the same animal, a clearer wording in sentence 3 of paragraph 1 of this section, would be '.....completely missing eyespots from the forewing and/or the hindwing...'

- **Change made.**

In the legend for Fig 2, part (c) shows both the dorsal and ventral forewing.

spalt disruption also produces extra sections of wing vein which can result in splitting of an eyespot (not unexpected from previous work). Table S2 also lists, in *Junonia*, 'vein-related discal spot distortion' but these (6 cases) are not mentioned - is this spot similarly split by extra vein material - this is intriguing as the discal spot is in a very different position on the wing and does it have central expression of *spalt* ?

- **We have included a picture of a discal spot disruption phenotype (Fig. S1e) and included this text in the Results: “In *J. coenia* we also observed several cases where vein abnormalities around the discal cell were associated with reduction of discal spot patterns (Fig. S1), however we tentatively speculate that these color pattern defects are due primarily to vein-related pattern disruption since *spalt* expression has not been observed in discal spot patterns.”**

The dramatic Vanessa findings for *distal-less* clearly indicate a contrasting repressive function in

eyespot development, despite its larval expression in the foci but not in corresponding positions in other wing-cells.

In the legend to Fig 3 part (b) the aberrant wing is said to show a 'strong deletion phenotype', which is rather odd wording!

- **We replaced “a strong deletion phenotype” with “several abnormalities”.**

The effects on *Junonia* are much weaker, showing no additional eyespots, few marginal effects and only one case of eyespot expansion (Fig 3 d). However, Fig 3e shows specimen JcM1 as an illustration of background pigmentation effects - does this wing not also show a major enlargement of the anterior eyespot (judging from the other wing and from Fig 2b)?

- **This is now noted in the figure legend and in Table S1 (along with several other more subtle presumptive phenotypes).**

These results are striking, somewhat variable between the two species and very surprising in the case of *distal-less* - all of these aspects are clearly and interestingly discussed in the paper.

This study is important and the intriguing results will certainly stimulate further work. The manuscript should certainly be published.

[Authors' response in bold]

REVIEWERS' COMMENTS:

Reviewer #1 (Remarks to the Author):

My earlier comments were fairly minor so I am happy with the manner in which the authors have responded to all comments. It is great that images are now available as supplementary material, however I was unable to match up the images with the data in Table S1 - the mutant IDs in the table did not seem to be associated with the images. This data needs to be better documented (there may be something I have missed, or it may be something about the way the images are presented on the website, but I suggest writing the individual IDs into the images in photoshop so they can be identified - otherwise its a bit pointless having these images available)

Otherwise fine to go ahead, I recommend publication

We have photoshopped specimen IDs directly onto the wing images as requested. We have also double-checked the correspondence between the reference table and the images to ensure accuracy.